# ZipVL: Efficient Large Vision-Language Models with Dynamic Token Sparsification and KV Cache Compression

## Abstract

The efficiency of large vision-language models (LVLMs) is constrained by the computational bottleneck of the attention mechanism during the prefill phase and the memory bottleneck of fetching the key-value (KV) cache in the decoding phase, particularly in scenarios involving high-resolution images or videos. Visual content often exhibits substantial redundancy, resulting in highly sparse attention maps within LVLMs. This sparsity can be leveraged to accelerate attention computation or compress the KV cache through various approaches. However, most studies focus on addressing only one of these bottlenecks and do not adequately support dynamic adjustment of sparsity concerning distinct layers or tasks. In this paper, we present ZipVL, an efficient inference framework designed for LVLMs that resolves both computation and memory bottlenecks through a dynamic ratio of important tokens. This ratio is adaptively determined based on the layer-specific distribution of attention scores, rather than fixed hyper-parameters, thereby improving efficiency for less complex tasks while maintaining high performance for more challenging ones. Then we select important tokens based on their normalized attention scores and perform attention mechanism solely on those important tokens to accelerate the prefill phase. To mitigate the memory bottleneck in the decoding phase, we employ mixed-precision quantization to the KV cache, where high-bit quantization is used for caches of important tokens, while low-bit quantization is applied to those of less importance. Our experiments demonstrate that ZipVL can accelerate the prefill phase by $2.6\times$ and reduce GPU memory usage by 50.0%, with a minimal accuracy reduction of only 0.2% on Video-MME benchmark over LongVA-7B model, effectively enhancing the generation efficiency of LVLMs.

## 1 Introduction

With the recent advancement of large language models (LLMs) (Achiam et al., 2023; Team et al., 2023; Vavekanand & Sam, 2024), many studies have extended their capabilities to comprehend and generate visual content. These models, commonly known as large vision-language models (LVLMs), have demonstrated remarkable performance in tasks such as image captioning and visual question answering (Ge et al., 2024b; Liu et al., 2024b; Team, 2024; Ge et al., 2024c; Lin et al., 2023). Typically, to remain compatible with the next-token-prediction generation scheme of LLMs, images or videos are encoded into visual tokens through a pre-trained visual encoder, and concatenated with text tokens for input into the model. For instance, LLaVA (Liu et al., 2024b) employs a pre-trained CLIP-ViT-L-336px model (Radford et al., 2021), which encodes an image of size $336\times336$ pixels to 576 visual tokens. However, for high-resolution images or videos, the visual encoder generates excessive sequences of visual tokens, significantly limiting the generative efficiency of LVLMs. Specifically, the prefill phase suffers from the quadratic complexity of the attention mechanism, resulting in **computational bottleneck** and prolonged time-to-first-token (TTFT). In the decoding phase, each new token interacts with all preceding tokens, requiring to fetch the full key-value (KV) cache from memory. This process slows down decoding due to **memory bottleneck**. Improving generative efficiency in both phases is essential for the practical deployment of LVLMs.

To address computational complexity in the prefill phase, sparse attention (Pagliardini et al., 2023; Jiang et al., 2024; Zhu et al., 2024) has emerged as an effective strategy, particularly suitable for

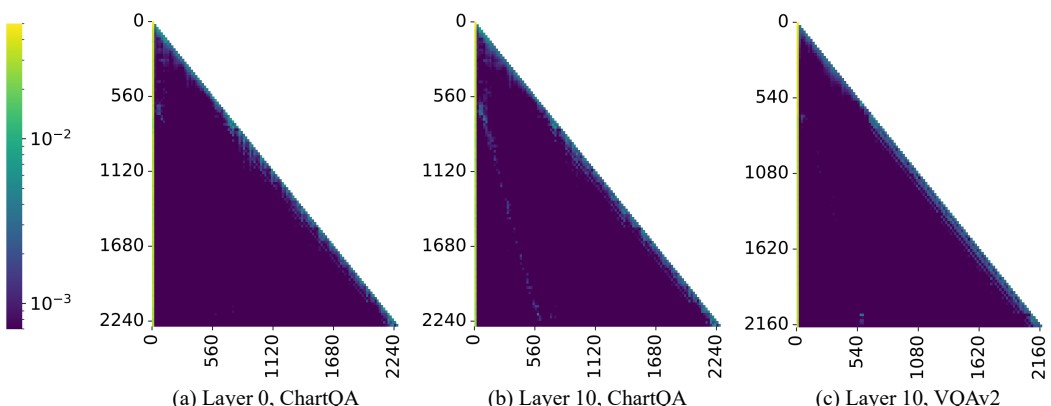

(a) Layer 0, ChartQA       (b) Layer 10, ChartQA       (c) Layer 10, VQAv2

Figure 1: The attention maps exhibit distinct sparse patterns across different layers (subfigures (a) and (b)) and vary significantly between tasks (subfigures (b) and (c)). Data was collected from the LLaVA-Next-7B model using input samples from the VQAv2 and ChartQA datasets.

LVLMs where visual information exhibits considerable redundancy, leading to highly sparse attention maps (Wan et al., 2024; Chen et al., 2024). This sparsity can be implemented at various levels of granularity. Some studies pre-define several sparse patterns and assign them to the attention mask during inference (Jiang et al., 2024; Zhu et al., 2024). However, these predefined patterns are not compatible with efficient attention implementations such as FlashAttention (Dao et al., 2022) and require custom GPU kernels for each pattern. Alternatively, other approaches adopt token-level sparsity by identifying and discarding less important tokens (Chen et al., 2024; Arif et al., 2024), allowing seamless integration with off-the-shelf efficient attention implementations. However, the optimal retention ratio of important tokens may vary across different layers or tasks due to distinct attention patterns, as illustrated in Figure 1. These methods rely on a fixed token retention ratio and do not dynamically adjust based on task difficulty, leading to suboptimal performance on complex tasks.

To alleviate memory bottleneck, various efforts have been made to reduce KV cache size, including token dropping (Wan et al., 2024), token merging (Yang et al., 2024a), and quantization (Hooper et al., 2024; He et al., 2024b). However, these methods often rely on fixed compression ratios that are uniformly applied across all layers, failing to account for the distinct characteristics of attention maps in different layers. Moreover, despite the necessity of identifying important tokens for both sparse attention and KV cache compression, a unified inference optimization framework has yet to be developed.

In this paper, we present ZipVL, an efficient inference framework tailored for LVLMs that jointly optimizes the prefill and decoding phases with a unified ratio of important tokens, as shown in Figure 2. To start with, we introduce a layer-wise adaptive ratio assignment scheme for important tokens. This ratio is adaptively determined based on the distribution of attention scores in each layer, rather than relying on predefined hyper-parameters (Chen et al., 2024; Arif et al., 2024; He et al., 2024b; Zhang et al., 2023). This adaptive approach allows the ratio to be adjusted according to task complexity, enhancing efficiency for simpler tasks while preserving performance for more complex ones. After determining the ratio, we then select important tokens with the highest normalized attention scores, following prior work (He et al., 2024b; Ren & Zhu, 2024). To alleviate the **computational bottleneck** in the prefill phase, sparse attention is performed at the token level by computing attention only for the selected important tokens. Notably, this approach seamlessly integrates with existing fast attention implementations without requiring custom GPU kernels. To tackle the **memory bottleneck**, the same set of important tokens is applied to compress the KV cache, where we employ high-bit quantization for caches of important tokens and low-bit quantization for those of less importance. Extensive experiments on multimodal benchmarks demonstrate that our method achieves nearly lossless performance while reducing prefill phase latency by $2.6\times$ and GPU memory usage by $50\%$.

In summary, our contributions are as follows:

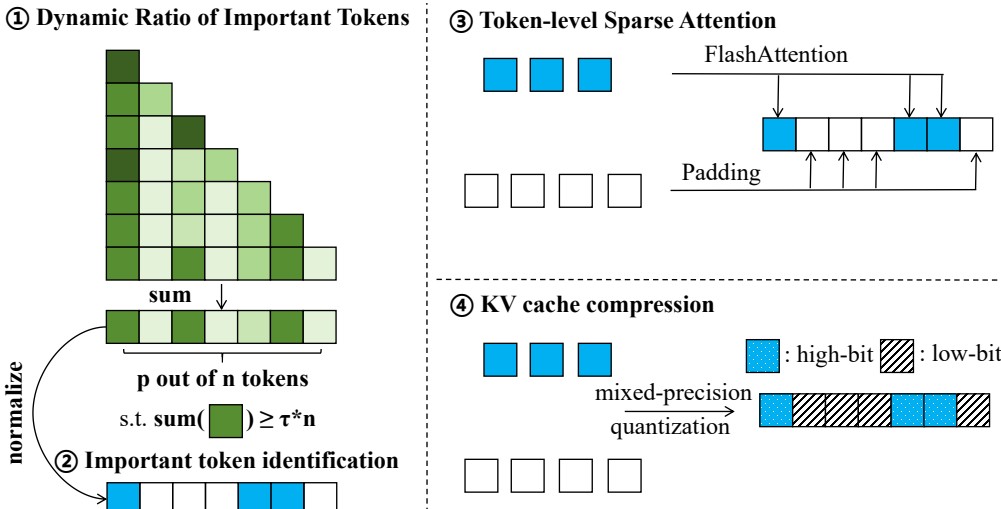

Figure 2: Overview of the proposed ZipVL framework during the prefill phase. Here, $\tau$ represents the threshold for retaining attention scores, $n$ and $p$ are the total number of tokens and the number of important tokens, respectively. After determining the ratio of important tokens and identifying them, we optimize the prefill phase by exclusively computing attention for important tokens. Additionally, we apply mixed-precision quantization to the KV cache, where the KV cache of less important tokens is quantized to a lower bit-width.

- We propose an adaptive layer-wise ratio assignment scheme for important tokens. The ratio is dynamically determined based on the distribution of attention scores and varies across different layers and tasks, thereby enhancing performance and efficiency compared to a fixed ratio scheme.

- We introduce a unified approach to jointly optimize the prefill and decoding stages through the assigned ratio of important tokens. Tokens considered less important are excluded from attention computation during the prefill phase, and their KV cache is quantized to a lower bit-width to alleviate the memory bottleneck in the decoding phase.

- By integrating these techniques, we present ZipVL, an efficient inference framework tailored for LVLMs. Comprehensive experiments across diverse benchmarks validate the efficacy of ZipVL, demonstrating that it achieves state-of-the-art performance in both accuracy and generation efficiency for LVLMs.

## 2 RELATED WORK

### 2.1 SPARSE ATTENTION FOR LLMS

Attention scores have been widely observed to exhibit high sparsity in both LLMs and LVLMs (Xiao et al., 2024; Wan et al., 2024; Zhu et al., 2024; Zaheer et al., 2020; Beltagy et al., 2020). This sparsity allows sparse attention to overcome the quadratic computational complexity of the standard attention mechanism by restricting each token to focus on only a subset of tokens within the input sequence (Zhu et al., 2024; Jiang et al., 2024; Pagliardini et al., 2023; Ribar et al., 2024). Depending on the granularity of sparsity, sparse attention can be categorized into unstructured, semi-structured, and structured schemes. The unstructured scheme (Lefaudeux et al., 2022; He et al., 2024a) employs sparse attention masks without a fixed structure, making it hardware-unfriendly and challenging to achieve practical inference acceleration. The semi-structured sparse attention uses attention masks with predefined sparse patterns (Jiang et al., 2024; Pagliardini et al., 2023; Zhu et al., 2024) or introduces N:M sparsity to attention weights (Chen et al., 2023). However, it requires customized computational kernels for each sparse pattern or specific hardware to achieve acceleration. Structured sparse attention (Chen et al., 2024; Arif et al., 2024) directly prunes tokens before the attention computation, enabling acceleration without the need for custom kernels. However, due to its coarse granularity, the pruning sparsity and the selection of tokens to prune significantly impact model

performance. For instance, HiRED (Arif et al., 2024) selects patches with the highest responses based on the feature maps of the visual encoder without considering the input text prompt, leading to suboptimal performance. FastV (Chen et al., 2024) empirically retains all tokens in the first two layers and prunes 50% of the visual tokens in all subsequent layers, resulting in performance degradation in challenging tasks such as ChartQA (Masry et al., 2022). In contrast, our approach achieves superior performance through an adaptive layer-wise ratio assignment scheme for important tokens.

## 2.2 KV CACHE COMPRESSION

KV cache prevents re-computation in the decoding phase by storing the key and value states of previous tokens, but with a significant memory bottleneck in long-context scenarios. Previous efforts to compress the KV cache can be broadly categorized into three types: token dropping-based (Ge et al., 2024a; Ren & Zhu, 2024; Zhang et al., 2023), token merging-based (Wang et al., 2024; Wan et al., 2024; Liu et al., 2024d), and quantization-based approaches (Hooper et al., 2024; He et al., 2024b; Yang et al., 2024b; Kang et al., 2024; Liu et al., 2024c). Both token dropping-based and merging-based methods aim to reduce the number of tokens stored in the KV cache by evicting or merging less important tokens. However, the information that is evicted or merged cannot be recovered, potentially leading to risks such as contextual incoherency or hallucination (Yang et al., 2024b), especially in multi-round dialogue scenarios. Conversely, quantization-based approaches retain all tokens in the KV cache and apply quantization to the cached values. To preserve performance, mixed-precision quantization further assigns higher bit-width to recent tokens (Liu et al., 2024c) or important tokens (Yang et al., 2024b) in the KV cache. In this paper, we apply mixed-precision quantization to compress the KV cache, leveraging the proposed layer-wise adaptive ratio assignment scheme to achieve a higher compression ratio.

## 3 PRELIMINARY

Attention block is the key module of Transformer-based LLMs. Each attention block contains three weight matrices $\mathbf{W}_Q, \mathbf{W}_K, \mathbf{W}_V \in \mathbb{R}^{d \times d}$, where $d$ is the dimension of the input data. Here, we use a single attention head and omit the output projection for clarity. In the prefill phase, the input data $\mathbf{X} \in \mathbb{R}^{n \times d}$ with a sequence length of $n$ is first multiplied with three weight matrices to obtain the query, key and value states:

$$\mathbf{Q} = \mathbf{X}\mathbf{W}_Q, \quad \mathbf{K} = \mathbf{X}\mathbf{W}_K, \quad \mathbf{V} = \mathbf{X}\mathbf{W}_V. \tag{1}$$

Then the attention output is calculated as follows:

$$\mathbf{A} = \mathrm{Softmax}\left(\frac{\mathbf{Q}\mathbf{K}^T + \mathbf{M}}{\sqrt{d}}\right), \mathbf{O} = \mathbf{A}\mathbf{V}. \tag{2}$$

Here, computing the product of $\mathbf{Q}\mathbf{K}^T$ has a quadratic complexity $O(n^2)$, which makes the prefill phase **compute-bound**. $\mathbf{M} \in \mathbb{R}^{n \times n}$ is a lower triangular causal mask to ensure that each token can only attend to itself and previous tokens. Unstructured and semi-structured sparse attention introduce sparsity in the attention mask $\mathbf{M}$ with dynamic or fixed sparse pattern. With custom computing kernels, tokens in certain positions can be skipped when computing $\mathbf{Q}\mathbf{K}^T$, thus accelerating the computation. On the other hand, structured sparse attention only computes attention scores for a subset of tokens $\mathbf{X}' \in \mathbb{R}^{n' \times d}$, reducing computational complexity to $O(n'^2)$ and seamlessly integrating with existing fast attention implementations.

For the decoding phase, the input data is the embedding of the current token $\mathbf{x} \in \mathbb{R}^{1 \times d}$. To enable the interaction between the current token and all previous tokens, the KV cache of previous tokens needs to be fetched from memory, making the decoding phase **memory-bound**:

$$\mathbf{q} = \mathbf{x}\mathbf{W}_Q, \quad \mathbf{K} = \mathrm{Concat}(\mathbf{K}, \mathbf{x}\mathbf{W}_K), \quad \mathbf{V} = \mathrm{Concat}(\mathbf{V}, \mathbf{x}\mathbf{W}_V). \tag{3}$$

The attention outputs are then computed as follows with a computational complexity of $O(n)$:

$$\mathbf{a} = \mathrm{Softmax}\left(\frac{\mathbf{q}\mathbf{K}^T}{\sqrt{d}}\right), \quad \mathbf{o} = \mathbf{a}\mathbf{V}. \tag{4}$$

# 4 METHOD

## 4.1 LAYER-WISE ADAPTIVE RATIO ASSIGNMENT FOR IMPORTANT TOKENS

Prior studies (Arif et al., 2024; Zhang et al., 2023; He et al., 2024b; Liu et al., 2024c; Wan et al., 2024) typically adopt a fixed ratio of important tokens across all layers. However, as analyzed by the preceding study (Chen et al., 2024) and demonstrated in Figure 1(a) and (b), there are substantial variations in the attention map patterns across different layers. Moreover, Figure 1(b) and (c) illustrate that, even within the same layer, attention maps can differ depending on the task and input. In scenarios involving complex tasks, a limited, static ratio for important tokens can impair model performance. This raises the question:

*can the model dynamically determine the number of tokens required to solve a task?*

Intuitively, for simpler tasks, the model needs to concentrate on fewer tokens, leading to a more focused distribution of attention scores. Conversely, more demanding tasks require the model to engage with a broader array of tokens, resulting in a more uniform distribution of attention scores. Prior work (Xiao et al., 2024) also highlights the criticality of preserving significant attention scores during inference within a constrained attention window. Building on these insights, we introduce a layer-wise adaptive scheme for assigning ratio of important tokens, ensuring the majority of significant attention scores are maintained within each layer.

Consider an attention layer with $n$ input tokens, where the full attention score matrix is denoted as $\mathbf{A} \in \mathbb{R}^{n \times n}$. The accumulated attention score for each token $j$ is calculated by summing the corresponding column:

$$a_j = \sum_{c=1}^{n} \mathbf{A}_{c,j}. \tag{5}$$

These accumulated attention scores are subsequently sorted in descending order, such that $a_{\text{sorted}(j)}$ represents the $j$-th highest attention score. The number of important tokens $p$ is determined by preserving the majority of attention scores with minimal number of tokens, which can be expressed as:

$$p = \min\{p \in \mathbb{Z} \mid \sum_{j=1}^{p} a_{\text{sorted}(j)} \geq \tau \times n\}. \tag{6}$$

Here, $\tau$ is the threshold dictating the retention of attention scores and the sum of the attention scores in $\mathbf{A}$ is equal to $n$ due to the row-wise $\text{Softmax}$ operation. As shown in Figure 3, our method can dynamically adjust the ratio of important tokens across distinct layers and tasks, thereby enhancing performance in complex tasks while improving efficiency in simpler tasks. Additional experimental results can be found in Section 5.2.1 and Figure 4.

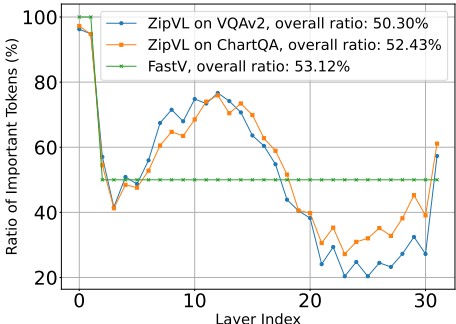

Figure 3: The ratio of important tokens distributed across layers. Data was collected from the LLaVA-Next-7B model using input samples from the VQAv2 and ChartQA datasets.

## 4.2 INFERENCE OPTIMIZATION WITH UNIFIED TOKEN RATIO

After determining the number of important tokens $p$ for each layer, we partition all tokens into two sets: set $\mathbf{T}$ of important tokens with a size of $p$ and the set $\mathbf{U}$ for less important tokens with a size of $n - p$. Following prior work (Ren & Zhu, 2024; He et al., 2024b), we use normalized attention scores to assess token importance, calculated as follows:

$$\tilde{a}_j = \frac{\sum_{c=1}^{n} \mathbf{A}_{c,j}}{\text{nnz}(\mathbf{A}_{:,j})}. \tag{7}$$

Here, $\mathrm{nnz}(\mathbf{A}_{:,j})$ denotes the number of non-zero elements in the $j$-th column. Important tokens are then selected using the top-$k$ indexing method, while the remainder are considered less important:

$$\mathbf{T} = \mathrm{topk\_index}(\tilde{a}_j, p), \tag{8}$$

$$\mathbf{U} = \{j \in \{1, 2, \ldots, n\} \mid j \notin \mathbf{T}\}. \tag{9}$$

The inference optimization is then performed based on the split of tokens. Specifically, to address the **computational bottleneck** in the prefill phase, the attention mechanism is performed solely on these important tokens, thereby enhancing efficiency through token-level sparsity. Tokens excluded from this computation have their outputs padded to maintain the number of tokens consistent for subsequent layers. By leveraging token-level sparsity, our approach seamlessly integrates with off-the-shelf, fast attention implementations (Dao et al., 2022) to expedite the prefill process.

To tackle the **memory bottleneck**, we implement mixed-precision quantization for the KV cache based on the same token split in Eqs. (8) and (9). The KV cache for important tokens is quantized at a higher bit-width to retain information, whereas the cache for less critical tokens is quantized at a lower bit-width to significantly reduce KV cache size. In comparison to prior method (He et al., 2024b), quantizing KV cache with our adaptive layer-wise token ratio leads to a higher compression ratio with even stronger performance. Further details will be provided in Section 5.3.1.

**Efficient approximation of full attention scores.** To integrate our method with fast attention implementation (Dao et al., 2022) and circumvent the computation of full attention scores in Eqs. (5) and (7), we selectively compute and accumulate the attention scores for a subset of tokens, following previous literature (He et al., 2024b; Jiang et al., 2024). The size of this subset is small and fixed, ensuring that the computational burden for these tokens remains minimal in long-context scenarios. The accumulated and normalized attention scores for each token can then be approximated with partial attention scores. Details can be found in Appendix A.

**Computation in the decoding phase.** It should be noted that sparse attention is exclusively utilized in the prefill phase to mitigate its computational bottleneck. During the decoding phase, the computation follows the standard attention mechanism as described in Eq. (3). Nonetheless, the KV cache for newly generated tokens will also be quantized with mixed precision. Specifically, every time 100 new tokens are generated, their importance is assessed based on the attention scores of the last token, and they are subsequently quantized accordingly.

Overall, the attention mechanism after optimization is summarized in Algorithm 1.

## 5 EXPERIMENTS

### 5.1 IMPLEMENTATION DETAILS

To assess the effectiveness of our proposed method, we conduct experiments on both image and video understanding tasks. For image understanding, we utilize three widely adopted LVLMs: LLaVA (Lin et al., 2023), LLaVA-Next (Liu et al., 2024a), and QWen-VL (Bai et al., 2023). These models are evaluated against five rigorous benchmarks: VQAv2 (Goyal et al., 2017), TextVQA (Singh et al., 2019), GQA (Hudson & Manning, 2019), MME (Fu et al., 2023), and ChartQA (Masry et al., 2022). For video understanding, evaluations are conducted using the LongVA (Zhang et al., 2024) model on the Video-MME (Fu et al., 2024) benchmark. To ensure reproducibility, all reported results are obtained using the Evaluation Suite of Large Multimodal Models (Li et al., 2024). For mixed-precision quantization, the KV cache of important tokens was quantized to 4-bit, while the KV cache of other tokens was quantized to 2-bit.

### 5.2 MAIN RESULTS

#### 5.2.1 EVALUATION ON IMAGE BENCHMARKS

We begin our evaluation on five image comprehension benchmarks and compare our results against well-established methods with token-level sparsity: FastV (Chen et al., 2024) and HiRED (Arif et al., 2024). The results are presented in Table 1. Notably, HiRED determines the importance of patches through the feature map of the visual encoder, without considering the semantic information of the input prompt, resulting in a significant accuracy drop. In contrast, both FastV and our approach assess

---

**Algorithm 1:** The attention mechanism of ZipVL

**procedure** `ZipVL Prefill`**:**

    **Input:** Input embedding $\mathbf{X}$, bit-width for important tokens $b_\text{h}$, bit-width for other tokens $b_\text{l}$

    **Output:** Attention output $\mathbf{O}$, KV cache $(\mathbf{K}, \mathbf{V})$

    Calculate query, key and value states $(\mathbf{Q}, \mathbf{K}, \mathbf{V})$ as per Eq. (1)

    Select a subset of tokens $\mathbf{Q}'$ from query states and compute attention scores

      $\mathbf{A}' = \text{Softmax}\left(\mathbf{Q}'\mathbf{K}^T\right)$

    Determine the number of important tokens as per Eq. (6)

    Calculate the normalized attention scores for each token as per Eq. (7)

    Select set $\mathbf{T}$ for important tokens as per Eq. (8) and set $\mathbf{U}$ for other tokens as per Eq. (9)

    `// Token-level Sparse Attention with FlashAttention`

    $\mathbf{O} = \text{FlashAttention}(\mathbf{Q}[\mathbf{T}], \mathbf{K}[\mathbf{T}], \mathbf{V}[\mathbf{T}])$

    `// Compressing KV Cache`

    $\hat{\mathbf{K}} = \text{Concat}(\text{Quant}(\mathbf{K}[\mathbf{T}], b_\text{h}), \text{Quant}(\mathbf{K}[\mathbf{U}], b_\text{l}))$

    $\hat{\mathbf{V}} = \text{Concat}(\text{Quant}(\mathbf{V}[\mathbf{T}], b_\text{h}), \text{Quant}(\mathbf{V}[\mathbf{U}], b_\text{l}))$

    **return** $\mathbf{O}$, $(\hat{\mathbf{K}}, \hat{\mathbf{V}})$

**procedure** `ZipVL Decoding`**:**

    **Input:** Input embedding $\mathbf{x}$, stored KV cache $(\mathbf{K}_\text{in}, \mathbf{V}_\text{in})$, bit-width for important tokens $b_\text{h}$,
        bit-width for other tokens $b_\text{l}$, the number of token generated $m$

    **Output:** Attention output $\mathbf{o}$, updated KV cache $(\mathbf{K}_\text{out}, \mathbf{V}_\text{out})$

    Calculate query, key and value states $(\mathbf{q}, \mathbf{k}, \mathbf{v})$ as per Eq. (1)

    Fetch KV cache from memory: $\mathbf{K} = \text{Concat}(\mathbf{K}_\text{in}, \mathbf{k})$,     $\mathbf{V} = \text{Concat}(\mathbf{V}_\text{in}, \mathbf{v})$

    Compute attention output $\mathbf{o} = \text{FlashAttention}(\mathbf{q}, \mathbf{K}, \mathbf{V})$

    `// Compress new KV cache every 100 tokens generated`

    **if** $m\%100 == 0$ **then**

        Compute attention scores: $\mathbf{a} = \text{Softmax}\left(\mathbf{q}\mathbf{K}^T\right)[-100:]$

        Determine the number of important tokens as per Eq. (6)

        Calculate the normalized attention scores as per Eq. (7)

        Select set $\mathbf{T}$ as per Eq. (8) and set $\mathbf{U}$ as per Eq. (9)

        $\mathbf{K}' = \mathbf{K}[-\mathbf{100}:], \mathbf{V}' = \mathbf{V}[-\mathbf{100}:]$

        $\hat{\mathbf{K}}' = \text{Concat}(\text{Quant}(\mathbf{K}'[\mathbf{T}], b_\text{h}), \text{Quant}(\mathbf{K}'[\mathbf{U}], b_\text{l}))$

        $\hat{\mathbf{V}}' = \text{Concat}(\text{Quant}(\mathbf{V}'[\mathbf{T}], b_\text{h}), \text{Quant}(\mathbf{V}'[\mathbf{U}], b_\text{l}))$

        $\mathbf{K}_\text{out} = \text{Concat}(\mathbf{K}[:-\mathbf{100}], \hat{\mathbf{K}}'), \mathbf{V}_\text{out} = \text{Concat}(\mathbf{V}[:-\mathbf{100}], \hat{\mathbf{V}}')$

    **else**

        $\mathbf{K}_\text{out} = \mathbf{K}, \mathbf{V}_\text{out} = \mathbf{V}$

    **return** $\mathbf{o}$, $(\mathbf{K}_\text{out}, \mathbf{V}_\text{out})$

---

token importance via attention maps in the LVLMs. However, FastV employs a fixed token ratio and exhibits severe performance degradation on challenging tasks such as ChartQA (Masry et al., 2022). By implementing layer-wise adaptive ratio assignment, our proposed ZipVL consistently surpasses FastV across all five benchmarks and three model architectures, while maintaining a smaller overall ratio of important tokens. As illustrated in Figure 4, our method dynamically adjusts the ratio across various tasks and models, slightly increasing the ratio of important tokens for difficult tasks to preserve performance and enhancing efficiency on simpler tasks. Moreover, the performance gap between our method and FastV becomes more pronounced over the LLaVA-Next-13B model. This discrepancy can be attributed to the varying attention maps across different models and that FastV's predefined hyperparameters are not universally applicable, whereas our dynamic approach demonstrates high robustness.

### 5.2.2 EVALUATION ON VIDEO BENCHMARKS

We also assess the performance of our method on the Video-MME benchmark (Fu et al., 2024) over the LongVA model (Zhang et al., 2024), which supports a maximum multimodal input length of $224K$ tokens. We compare our approach with semi-structured sparse attention methods such as MInference (Jiang et al., 2024) and QK-sparse (Pagliardini et al., 2023), as well as the structured sparse attention

Table 1: Performance comparisons of image LVLMs on various benchmarks. Here, "Ratio" denotes the proportion of tokens participating in attention computation. "†" denotes token-level sparsity is only employed in attention modules.

| Model | Method | Ratio | VQAv2 | ChartQA | TextVQA | GQA | MME |
|---|---|---|---|---|---|---|---|
| LLaVA-v1.5-7B | Full | 100% | 76.6 | 18.2 | 46.1 | 61.9 | 1507 |
| | FastV† | 53.1% | 75.8 | 17.7 | 45.5 | 60.2 | 1511 |
| | HiRED | 20% | 73.0 | 17.3 | 45.6 | 56.8 | 1368 |
| | HiRED | 40% | 75.5 | 17.6 | 45.6 | 59.5 | 1433 |
| | Ours ($\tau$=0.96) | 44.1% | 76.1 | 17.9 | 45.0 | 61.3 | 1515 |
| | Ours ($\tau$=0.975) | 52.8% | **76.4** | **18.0** | **45.7** | **61.7** | **1524** |
| LLaVA-Next-7B | Full | 100% | 80.3 | 54.8 | 64.8 | 64.1 | 1519 |
| | FastV† | 53.1% | 79.5 | 51.2 | 63.7 | 63.7 | 1490 |
| | HiRED | 20% | 77.5 | 42.0 | 61.4 | 61.4 | 1483 |
| | HiRED | 40% | 78.8 | 46.5 | 61.8 | 59.4 | 1474 |
| | Ours ($\tau$=0.96) | 40.4% | 79.4 | 51.0 | 62.6 | 63.8 | 1489 |
| | Ours ($\tau$=0.975) | 49.7% | **79.8** | **52.4** | **63.9** | **64.1** | **1495** |
| LLaVA-Next-13B | Full | 100% | 80.9 | 66.2 | 66.9 | 65.7 | 1570 |
| | FastV† | 53.1% | 76.8 | 51.6 | 59.7 | 62.9 | 1555 |
| | HiRED | 20% | 77.9 | 48.9 | 63.6 | 63.1 | 1545 |
| | HiRED | 40% | 79.3 | 53.7 | **65.2** | 64.1 | **1570** |
| | Ours ($\tau$=0.96) | 30.6% | 79.7 | 56.2 | 63.8 | 64.4 | 1549 |
| | Ours ($\tau$=0.975) | 36.7% | **80.3** | **58.2** | 65.0 | **65.0** | 1551 |

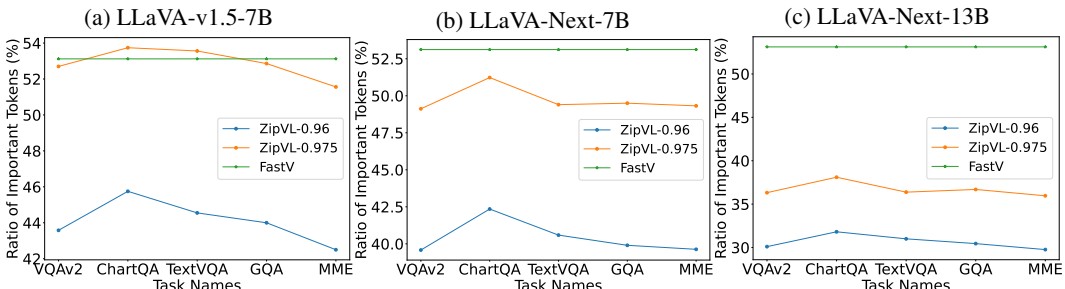

Figure 4: The ratio of important tokens across different methods on different tasks. The proposed ZipVL can adaptively determine this ratio based on the attention scores, assigning more ratio to important tokens on complex tasks.

method FastV (Chen et al., 2024). The results are summarized in Table 2. Among these sparse attention methods, FastV (Chen et al., 2024) consistently retains a fixed proportion of tokens while MInference (Jiang et al., 2024) retains a fixed number of sparse blocks. Notably, our approach not only achieves the highest overall performance but also exhibits superior reductions in FLOPs within the attention module compared to other sparse attention methods. This demonstrates the effectiveness of employing dynamic token-level sparsity to accelerate the attention module in LVLMs. Furthermore, long videos inherently contain significant redundancy, and our method dynamically allocates the ratio of important tokens by analyzing the sparse attention maps, resulting in a higher FLOPs reduction ratio when processing 128-frame videos compared to 64-frame videos.

## 5.3 ABLATION STUDY

### 5.3.1 EFFECT OF THE LAYER-WISE ADAPTIVE RATIO

In this subsection, we evaluate the efficacy of the proposed adaptive ratio assignment scheme by integrating it with sparse attention and KV cache compression, as detailed in Table 3. Initially, we implement a fixed sparse attention scheme on LongVA-7B model over Video-MME benchmark. In this scheme, the ratio for important tokens remains constant across all attention layers and is fixed.

Table 2: Performance comparisons of video LVLMs on Video-MME benchmark. Here, "Attn FLOPs Reduction" denotes the reduction in floating-point operations (FLOPs) of the attention mechanism. "†" denotes token-level sparsity is only employed in attention modules.

| Model | Frames | Method | Attn FLOPs Reduction | Short | Medium | Long | Overall |
|-------|--------|--------|----------------------|-------|--------|------|---------|
| LongVA-7B | 64 | Full | 0% | 61.4 | 50.9 | 45.0 | 52.4 |
| | | QK-sparse | 47.0% | 60.9 | 51.4 | **45.1** | 52.4 |
| | | MInference | 54.2% | 60.7 | 51.2 | 44.6 | 52.1 |
| | | FastV† | 71.7% | 61.0 | 50.6 | 45.0 | 52.2 |
| | | Ours($\tau$=0.975) | 77.0% | **61.1** | **51.6** | 45.0 | **52.5** |
| | 128 | Full | 0% | 61.1 | 50.4 | 46.2 | 52.6 |
| | | QK-sparse | 46.9% | **61.3** | 49.7 | **46.3** | 52.4 |
| | | MInference | 77.1% | 61.0 | 50.5 | 45.3 | 52.3 |
| | | FastV† | 71.7% | 60.2 | 50.2 | 46.2 | 52.2 |
| | | Ours($\tau$=0.975) | 82.3% | 60.7 | **51.3** | 45.2 | **52.4** |

Although this approach shares the same overall important token ratio and FLOPs reduction ratio as our method, it suffers from significant performance degradation (51.1% vs. 52.6%) due to its failure to account for the varying attention maps across layers. In contrast, our method achieves nearly lossless performance (52.4% vs. 52.6%) while reducing the FLOPs of attention mechanism by 82.3%.

To assess the efficacy of our method specifically for KV cache compression, we apply it to compress the KV cache of LLaMA3-8B (Meta, 2024) and evaluate its performance on the GSM8k dataset. The baseline method (He et al., 2024b) also utilizes mixed-precision quantization for KV cache but employs a fixed ratio for important tokens. For both the baseline and our method, important tokens are quantized to 4-bit, while other tokens are quantized to 2-bit. Notably, by adaptively determining the ratio of important tokens, our method achieves a significantly higher compression ratio (6.18 × vs. 4.69×) while maintaining superior accuracy (54.06% vs. 53.75%). This demonstrates that **our method also sets a new state-of-the-art for KV cache compression of LLMs**.

Table 3: The effect of the proposed adaptive ratio assignment scheme on sparse attention and KV cache compression. Here, "Ratio" denotes the proportion of important tokens. For Video-MME benchmark, the input videos consist of 128 frames.

| **Sparse Attention** | | | |
|----------------------|-----------|----------------------------|----------------|
| Method | Ratio (%) | Attn FLOPs Reduction (%) | Video-MME (%) |
| LongVA-7B | 100 | 0 | 52.6 |
| Fixed | 42.1 | 82.3 | 51.1 |
| Ours | 42.1 | 82.3 | **52.4** |
| **KV Cache Compression** | | | |
| Method | Ratio (%) | Compression Ratio | GSM8k Acc. (%) |
| LLaMA3-8B | 100 | 1× | 55.88 |
| Fixed (He et al., 2024b) | 70.0 | 4.69× | 53.75 |
| Ours | **28.6** | **6.18×** | **54.06** |

### 5.3.2 Effect of the threshold $\tau$

We further investigate the impact of the attention retention threshold $\tau$ on both the ratio of important tokens and model performance. The results are illustrated in Figure 5. Intuitively, a lower retention threshold leads to a reduced ratio for important tokens, thereby enhancing generation efficiency at the cost of performance degradation. Notably, the ratio decreases significantly as $\tau$ decreases but remains above 0.97, with minimal performance deterioration. Conversely, when $\tau$ falls below 0.97, substantial performance loss is observed, despite a gradual reduction in the ratio of important tokens. This indicates that the optimal range for $\tau$ lies around 0.97.

### 5.4 Deployment Efficiency

In this subsection, we present the prefill phase latency and GPU memory usage in Figure 6 to illustrate the real efficiency improvements achieved by ZipVL. Specifically, we first compare the prefill phase latency of ZipVL with that of the well-established semi-structured sparse attention method, MInference (Jiang et al., 2024), as shown in Figure 6a. Notably, MInference exhibits significant additional overhead when the sequence length is short and is notably slower than FlashAttention (Dao et al., 2022) for sequence lengths below 32K. In contrast, ZipVL achieves comparable latency to FlashAttention with short input sequences, while significantly reducing the prefill phase latency as the sequence length exceeds 32K. This can be attributed to the fact that the attention module's latency becomes the dominant factor in the total latency with long sequences. With an input sequence length of 200K, ZipVL achieves a 2.6× reduction in prefill-phase latency.

Moreover, MInference is not designed to reduce the KV cache size, while the proposed ZipVL jointly optimizes the attention computation in the prefill phase and the KV cache through the dynamic token ratio. Consequently, ZipVL presents a 50.0% reduction in GPU memory usage with an input sequence length of 64K (26,230MB vs. 52,514MB), as illustrated in Figure 6b.

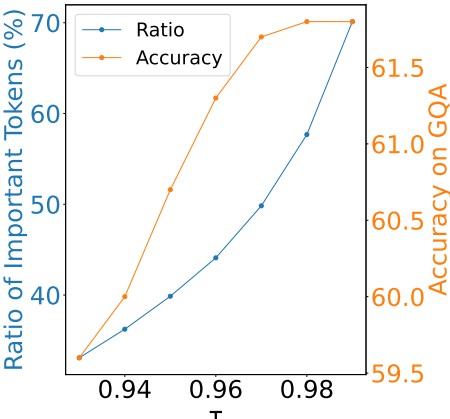

Figure 5: The effect of attention scores retention threshold $\tau$ on the ratio of important tokens and the model performance. Data was collected on GQA benchmark over LLaVA-v1.5-7B model.

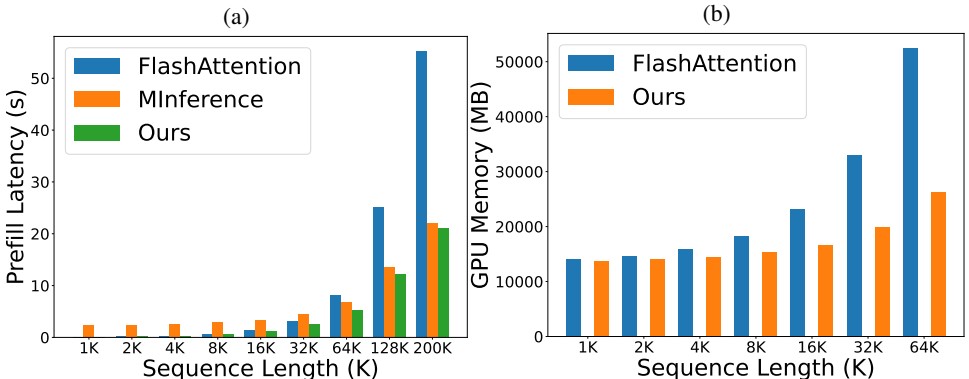

Figure 6: Comparisons of prefill phase latency and GPU memory across different sequence lengths. Data is collected from LongVA-7B model.

## 6    CONCLUSION AND FUTURE WORK

In this paper, we have proposed ZipVL, an efficient inference framework tailored for LVLMs. ZipVL jointly optimizes both the prefill and decoding phases by assigning an adaptive ratio of important tokens. This ratio is dynamically adjusted based on the distribution of attention scores across each layer, ensuring that the majority of attention scores are preserved. After identifying important tokens through normalized attention scores, less significant tokens are excluded from attention computation during the prefill phase to alleviate the computational bottleneck. Additionally, their KV cache is quantized to a lower bit-width, mitigating the memory bottleneck in the decoding phase. Extensive experiments have demonstrated that ZipVL significantly enhances the generation efficiency of LVLMs, achieving up to a 2.6× reduction in prefill phase latency and a 50% reduction in GPU memory usage. However, a limitation of our approach is its focus on sparse attention during the prefill phase only, while attention during the decoding phase and the multi-layer perceptron (MLP) modules in both phases remain dense. Future efforts may explore extending sparse computations to MLP modules or the attention mechanism in the decoding phase to further reduce computational complexity.

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

# Appendix

## A EFFICIENT APPROXIMATION OF FULL ATTENTION SCORES

ZipVL requires accumulated attention scores to adaptively assign the ratio of important tokens and normalized attention scores to identify token importance. However, attention scores are not accessible in fast attention implementations such as FlashAttention (Dao et al., 2022). To integrate our method with FlashAttention, we follow prior literature (He et al., 2024b; Jiang et al., 2024) and select a subset of tokens, referred to as "probe tokens" (He et al., 2024b), and explicitly compute their attention scores:

$$\mathbf{A}_{probe} = \text{Softmax} \left( \frac{\mathbf{Q}_{probe}\mathbf{K}^T}{\sqrt{d_k}} \right).\qquad(10)$$

The approximate accumulated and normalized attention scores for each token can then be obtained accordingly based on $\mathbf{A}_{probe}$. Prior work (He et al., 2024b) selects 10% of the tokens as probe tokens, which still yields quadratic complexity in Eq. 10. In contrast, we select only 64 recent tokens and 64 randomly positioned tokens, which incurs negligible computation overhead in long-context scenarios.

