# OpenReview forum: "ZipVL: Efficient Large Vision-Language Models with Dynamic Token Sparsification and KV Cache Compression"
_ICLR.cc/2025/Conference — ICLR 2025 Conference Withdrawn Submission_

### Official Review · Reviewer_TjhL · 2024-11-03

**Soundness:** 2
**Presentation:** 3
**Contribution:** 2
**Rating:** 5
**Confidence:** 3

**Summary:**

The paper introduces a comprehensive LVLM inference framework that reduces latency of prefill phase and memory utilization of decode phase by using a novel attention score based filtering of tokens. The attention scores of a subset of tokens from query states are ranked and are used to identify the number of important tokens required to meet a certain threshold. An attention matrix with approximate attention scores are used to select the most important tokens. A subset of the Q and K matrices with the selected important tokens are used for flash attention computation. The importance of the tokens also dictates their precision in the KV cache enabling a mixed precision quantization for efficient memory utilization. The ratio of the important tokens is identified for every layer dynamically based on the input tokens. The efficiency of the proposed scheme is demonstrated by evaluating a few image LVLMs and a video LVLM model across various benchmarks and compared against other related works. The proposed method cuts down prefill phase latency by as much as 2.6x and reduces GPU memory usage by 50% while achieving good model performance.
The original contribution of this paper is the adaptive layer-wise important token ratio identification scheme to overcome the drawbacks of fixed ratio method and demonstrating its effectiveness in cutting down on prefill-latency and memory utilization without much model performance degradation.

**Strengths:**

The motivation for a layer-wise adaptive ratio scheme is presented well. The evaluation section has a good mix of models and benchmarks evaluated across other recent works. The results section demonstrates the proposed scheme's superiority to fixed ratio methods in terms of model accuracy. The related works sections discuss the most recent works and highlight the drawbacks in their methods.

**Weaknesses:**

The paper lacks any quantitative evaluation of the overheads of the proposed adaptive ratio scheme. The paper mentions the overhead of the adaptive ratio computation to be negligible for large context lengths but how does it fair for smaller context lengths? Are fixed ratio schemes better for lower context lengths?
Fig 6a compares the prefill latency with MInference and Flash attention but another comparison point for a fixed ratio scheme would be insightful.
Similarly, the latency information (at least for Full, Fixed/FastV, and adaptive methods) for benchmarks in Table-1,2,3 would help understand if the better model accuracy is at the expense of increase in runtime.
It's also not clear how 64 recent + 64 random tokens are sufficient to approximately represent long context scenarios.

**Questions:**

How are the attention scores for all the tokens computed from the attention scores of the probe tokens?
How are 64 recent and 64 randomly chosen tokens sufficient for large context length models? I see that your scheme has better model accuracy than FastV but why is this the case? Are 128 probe tokens really sufficient to rebuild the full approximate attention matrix of large sequences?
What kind of GPUs were used for the experiments?

---

### Official Review · Reviewer_GKoA · 2024-11-03

**Soundness:** 2
**Presentation:** 2
**Contribution:** 2
**Rating:** 3
**Confidence:** 4

**Summary:**

This paper proposes an inference framework for LVLMs that resolves both the computation and memory bottleneck by dynamically selecting the important tokens for each layer. The important token selection is based on each token's average attention score, and the less important tokens are padded out in the following layers. In the prefill phase, the attention mechanism is performed solely on the selected tokens for each layer, thus reducing computation. In the decoding stage, only the KV cache for the important tokens is quantized in higher bit-width, effectively reducing memory usage while maintaining high accuracy. The evaluations show that this work exhibits competitive performance in terms of accuracy and speed compared to baseline methods.

**Strengths:**

1.	Improving end-to-end LVLM inference is an important and urgent issue.
2.	The motivation for using adaptive token selection per layer is solid.
3.	Evaluations show promising results.

**Weaknesses:**

1. The proposed method needs more detailed explanations. A few points that needs clarification are

	i. The overhead of important token selection, which seems quite large

	ii. How hyperparameters are chosen, such as τ and the KVcache compression bit size (4bit/2bit)

2. Evaluations has room for improvement. Some experiments I feel are missing are

	i. Breakdown of execution time, which helps visualize where the improvements came from.

	ii. More sensitivity tests on τ, on different models and datasets

	iii. Sensitivity test on KV cache compression bit size

	iv. KV cache compression results on LVLM models, not LLM models

3. The KV cache compression method is not noble, as the identical method (metric to determine important token, selectively compressing important tokens to higher bit-width) was already proposed in [1]. The only difference between the two is the important token selection ratio.

[1] ZipCache: Accurate and Efficient KV Cache Quantization with Salient Token Identification, arXiv:2405.14256, 2024

**Questions:**

Thank you for your submission. I enjoyed reading this paper, and have some follow-up comments and questions related to the comments in the weakness section.

1. The layer-wise adaptive ratio assignment relies on the hyperparameter τ, yet the evaluations analyze τ based on only one model, with one dataset. Will τ show similar tendencies with other models / datasets?

2. What would happen if the proposed method were to be applied to larger, deeper models? My main concern is that pruning tokens on every layer in deeper models will lead to minimal amount of tokens being left in the final layers, which may lead to accuracy degradation.

3. The proposed KV cache compression method should be tested on LVLM models too, as their applicability and performance on LVLM models is the main proposal of this work. How much accuracy drop will this incur?

4. From what I understand, the ratio of tokens participating in attention computation is different for every layer in this work. Then how is the ratio in Table 1 computed for this work? Is it the average number of participating tokens?

5. From the results in table 3, this work seems to fit in well with LLMs as well. why is the target system limited to LVLMs?

---

### Official Review · Reviewer_f4GA · 2024-11-04

**Soundness:** 2
**Presentation:** 2
**Contribution:** 2
**Rating:** 3
**Confidence:** 5

**Summary:**

This paper proposes a novel inference framework, ZipVL, to address the computational and memory bottlenecks of LVMs. The framework dynamically determines the ratio of important tokens based on layer-specific attention distributions. Key tokens are then sampled according to normalized attention scores to accelerate the prefill phase. During decoding, mixed-precision quantization is applied to the KV cache. Empirical experiments demonstrate ZipVL’s superior performance in terms of both accuracy and memory efficiency.

**Strengths:**

This paper is well-written and easy to follow. It applies dynamic token sparsification and mixed-precision quantization to LVLMs, highlighting the significance of KV cache compression for these models. Specifically, it explores more fine-grained token ratio allocation across layers, enhancing decoding efficiency.

**Weaknesses:**

1. Although ZipVL is specifically designed for LVLMs, the unique attention properties of LVLMs compared to LLMs—such as distinct attention mechanisms for vision tokens versus text tokens—are not sufficiently detailed. Prior work, including PyramidInfer and PyramidKV, also emphasizes layer-wise sparsity in LLMs. It would be helpful to clearly outline these distinct contributions.
2. In section 5, the baselines are not strong. Please include PyramidInfer or PyramidKV as baselines to strengthen comparisons, as they also explore layer-wise ratio assignments.
3. Section 3 currently focuses on LLM preliminaries; however, for LVLMs, it would be more relevant to discuss how vision tokens are concatenated with text tokens during prefill and how this integration influences decoding.
4. In the introduction, this method is described as "selecting important tokens with the highest normalized attention scores." While previous work shows that normalized attention scores are effective in LLMs, it's uncertain if this holds true for VLMs. An discussion on this topic would enhance the clarity of your argument.
5. Applying mixed-precision quantization during decoding seems intuitive, and substantial prior work—such as ZipCache, KVQuant, Kivi, and No Token Left Behind—has explored quantizing the KV cache to accelerate decoding. It would be beneficial to clarify the novelty of this approach and provide a comparison to existing methods.

**Questions:**

1. For efficiently approximating full attention scores, how can it be ensured that the insights gained from the selected, computed attention scores are reliable?
2. In Section 4, regarding computation in the decoding phase: if only the attention score of the last token is used for token evaluation, could this approach risk dropping key tokens unexpectedly?

---

### Official Review · Reviewer_ZNps · 2024-11-11

**Soundness:** 3
**Presentation:** 3
**Contribution:** 3
**Rating:** 5
**Confidence:** 3

**Summary:**

This paper describes a transformer model inference algorithm for reducing computation using sparsity (during prefill) and memory overheads using KV-cache compression applied to large vision language models.  The scheme proposes to enable a varying degree of sparsity and mixed-precision compression based upon an estimate of token importance using attention scores.   In the evaluation important tokens are quantized to 4-bits and the remaining tokens are quantized to 2-bits.  The evaluation shows limited degradation in performance on a range of question answering benchmarks.  Evaluation shows latency and memory consumption improvements during prefill latency and provides some sensitivity analysis of the main hyper parameter employed when measuring token importance.

**Strengths:**

Tackling an important problem: system bottlenecks in LLMs and LVLMs.
Presented results demonstrate the proposed approach works.

**Weaknesses:**

Somewhat limited novelty as sparsity is a well known technique for transformers and mixed-precision quantization has been applied to KV-cache before (the novelty is in how these are set).

Missing an evaluation on impact of the proposal on decoding latency/throughput.  Unclear if overhead of compression is accounted for in measurements.

Differences between Figure 1(a), (b) and (c) are not that apparent (at least visually) as the dark blue areas swamp the entire figure. Perhaps by applying some sort of threshold function to the data you can make the changes more apparent.

Adds two hyperparameters: $\tau$ used to select which tokens are important and decoding compression interval (set to 100 in Algorithm 1).  While there is some evaluation of the impact of $\tau$ the impact of varying the compression interval is not reported.

Unclear how Equations 6, 7 and 8 are evaluated (on CPU or GPU) and what the overheads are.  Can you please specify where these equations are evaluated and provide details on their computational overhead?

Very limited supplemental material (beyond code) provided.

**Questions:**

Why isn't the normalization in Equation 7 also applied in Equation 6?

What value of $\tau$ was used when obtaining the data in Figure 3?

Figure 5 shows the impact of $\tau$ on GQA for LLaVA-v1.5-7B, but what is unclear is if the values that work there would work for other networks and tasks.  Can you comment and/or provide additional sensitivity results for other networks/tasks?  Why isn't $\tau$ listed as an input in both procedures in Algorithm 1 (it is used in Equation 6 inside both)?

The value 100 used between line 347 and 348 in Algorithm 1 seems to be a hyperparameter (which could may be called something like a decoding compression interval).   How do results vary as this hyperparameter changes?   Beyond one sentence at the end of Section 4, I did not see any discussion of the process involved (compressing the KV cache every 100 generated tokens).

Does the prefill latency in Figure 6 include time to compress the K-V cache?  Assuming it is, what fraction of prefill latency is taken up by the compression phase?

What is the impact of the K-V cache compression on decoding latency and/or throughput?

What is the impact of the decoding compression interval on performance and decoding latency/throughput?

I was confused at Lines 275-277 about the following:  How can one "perform" the "attention mechanism" "solely" on important tokens before first having computing the attention scores required to determine token importance (Equations 6 and 7)?   Suggest you clarify the order of operations and explain how token importance is determined before applying the attention mechanism.  It seems to me that more specifically what is being saved is overhead on softmax and the MLP layers (I assume this  is what "FlashAttention()" on Line 335, 346 computes) but the calculations of Equation 6, 7 and 8 must introduce some computation overheads and there does not seem to be much said about the overheads in the paper.  For example I think Equation 6 and 8 would involve sorting, which may be slow.   Are Equations 6, 7 and 8 evaluated on CPU or GPU?  How much overhead is introduced and is that included in your latency measurements in Figure 6?

---

### Note · Authors · 2024-11-15

I have read and agree with the venue's withdrawal policy on behalf of myself and my co-authors.